# Elastic Provisioning of Network and Computing Resources at the Edge for IoT Services

**DOI:** 10.3390/s23052762

**Published:** 2023-03-02

**Authors:** Patrícia Cardoso, José Moura, Rui Neto Marinheiro

**Affiliations:** 1Instituto Universitário de Lisboa (ISCTE-IUL), Av. das Forças Armadas, 1649-026 Lisboa, Portugal; 2Departamento de Ciências e Tecnologias da Informação, Instituto de Telecomunicações, Instituto Universitário de Lisboa (ISCTE-IUL), Av. das Forças Armadas, 1649-026 Lisboa, Portugal

**Keywords:** resource management, elastic provisioning, software-defined networking, internet of things, edge computing, container, self-activation, self-release, fog computing, scarce resources

## Abstract

The fast growth of Internet-connected embedded devices demands new system capabilities at the network edge, such as provisioning local data services on both limited network and computational resources. The current contribution addresses the previous problem by enhancing the usage of scarce edge resources. It designs, deploys, and tests a new solution that incorporates the positive functional advantages offered by software-defined networking (SDN), network function virtualization (NFV), and fog computing (FC). Our proposal autonomously activates or deactivates embedded virtualized resources, in response to clients’ requests for edge services. Complementing existing literature, the obtained results from extensive tests on our programmable proposal show the superior performance of the proposed elastic edge resource provisioning algorithm, which also assumes an SDN controller with proactive OpenFlow behavior. According to our results, the maximum flow rate for the proactive controller is 15% higher; the maximum delay is 83% smaller; and the loss is 20% smaller compared to when the non-proactive controller is in operation. This improvement in flow quality is complemented by a reduction in control channel workload. The controller also records the time duration of each edge service session, which can enable the accounting of used resources per session.

## 1. Introduction

The International Data Corporate estimates that at the end of 2025 there will be 41.6 billion things connected to the Internet [1], generating 79.4 zettabytes of data. These predictions are supported by two evident pillars: (i) improvements in the Telecom sector, with more ubiquitous and cheaper Internet access, and (ii) programmable open-source solutions that make it possible to deploy very versatile network devices, aggregating networking, processing, and storage capabilities at the network edge.

The enormous scale of connected devices and the management of very large amounts of generated data, as well as the use of diverse communication protocols, hardware, and software supplied by distinct manufacturers, make it difficult to find the most efficient configuration to use in legacy network architectures, and to address emerging internet of things (IoT) scenarios. As such, there is a requirement for a more flexible and comprehensive network architecture, essentially at the network edge.

The most promising approach to address the challenges mentioned above is to add an edge layer between local system-embedded devices and the cloud to support peripheral computation, communication, and data storage. This new layer can enhance service performance, mobility support, and data privacy while reducing data volume exchanged through the backhaul links and guaranteeing low latency for end-user accessing services [2]. However, the addition of an edge layer to the system architecture brings new challenges in managing the available resources at the peripheral infrastructure [3,4] and associated services, which could be offered in a federated way [5].

The available edge computational resources, sometimes referred to as fog computing (FC), should be managed differently than computing assets from remote cloud data centers. This is because, among other issues, edge servers have fewer processing resources available when compared to cloud servers [3]. In this paper, we consider FC and edge computing (EC) concepts in a similar way to model scenarios with computational containers operating at network peripheral domains. Nevertheless, the reader is referred to [6], which discusses FC and EC as slightly distinct but relevant pillars of the current evolution of networked systems to satisfy the major requisites of IoT services.

Software-defined networking (SDN) has emerged as a network design to catalyze the real deployment of innovative control or management solutions, overcoming the network ossification imposed by the legacy proprietary network services [7]. In traditional networks, the control and the data planes are located within the network devices, requiring a personalized configuration on each device using low-level, rigid, and often vendor-specific commands. SDN takes the control out of the data plane devices and provides centralized logical control and abstraction from hardware complexity. Thus, SDN-based solutions provide automatic control loops for a stable and optimal system operation, using innovative and open-source programmable networking services [8].

Additionally, the use of virtualization techniques applied to SDN and sensor services can address scalability and heterogeneity issues imposed by most upcoming IoT-based scenarios [9]. Virtualized sensors abstract the hardware complexity from the developed software, reducing the number of physical devices, aggregating IoT data, and enabling the management of IoT domains via well-defined application programming interfaces (APIs) [10,11].

Considering all the previously discussed aspects, we found a strong motivation to enhance the current available literature by designing, implementing, and evaluating a novel programmable open-source solution, which optimizes the usage of limited edge resources and protects the quality of data flows. Summing up, the new key contributions of this work are as follows: (i) supporting virtualized resource management at the network edge; (ii) liberating idle system resources, ensuring a more efficient operation of the system’s limited resources; (iii) enhancing the quality of data flows by using a proactive SDN controller; and (iv) accounting resources used by each user service session, by the controller, enabling the exploration of novel dynamic business models at the network edge.

The paper structure is as follows. Section 2 revises the most relevant related literature, highlighting the novelty of our work. Section 3 presents the design and implementation details of the proposed solution. The evaluation results of this proposal are discussed in Section 4. The major outputs of the current investigation are summarized in Section 5. Finally, Section 6 concludes and points out some future research directions.

## 2. Related Work

This section discusses related work, highlighting the novelty of the present research. The text bellow debates data-driven softwarized solutions that are relevant for emerging data-intensive and time-sensitive scenarios, which help to achieve various goals such as controlling, orchestrating, or abstracting the available network edge assets, mainly for distributed computing resources. Table 1 compares the current work with previous associated research, highlighting the novel aspects of our contribution.

The work in [12] discussed adaptable and data-driven decision making for communication systems. They proposed machine learning (ML) [13] modules to enhance the functional primitives of observation (ensured mainly via SDN), composition (ensured mainly via network function virtualization—NFV), and control (ensured by the coordination between SDN and NFV) in the presence of uncertainty in relation to the network status evolution. By offering these enhancements, the data-driven networked systems can learn and properly react to changes in the networking context as well as unexpected variations in traffic load. In addition, [14] proposed a smart SDN management of fog services. The work in [15] studied the orchestration of SDN applications that cooperate to offer network-wide resilience. The studied applications involved traffic classification, anomaly detection, and traffic shaping.

Kobo et al. [10] studied SDN as a technology enabler for the upcoming use cases of wireless sensor networks. The work in [16] investigated SDN for IoT applications. They combined SDN with virtualization frameworks, such as NFV and network slicing. NFV can use high-level policies to manage IoT resources and network slicing can support flow-based QoS/QoE differentiation at the network edge. The authors of [17] studied the synergies of both SDN and NFV for the efficient and secure operation of IoT networks.

Tomovic et al. [18] designed a solution that combines the major benefits normally offered by both SDN and FC. Their proposal orchestrates fog resources, via SDN controllers, to diminish the level of complexity to efficiently control those resources. In addition, the SDN scalability issue was tamed by delegating some controller’s processing tasks to fog computing nodes. The virtualization was out of scope of their work.

Another approach involving SDN and container-based technology is the one described by Xu et al. [19], who proposed an in-house controller for elastically managing Docker computing resources at edge switches. They used an SDN controller pushed to the edge to manage the life cycle of services. Our research is also aligned with the main objective discussed in [20], which adopted localized flow management performed by distributed controllers to overcome the additional delay imposed by the constraints on the control channel between each SDN controller and the SDN-based switch. However, the Xu proposal was deployed in the Docker platform and was heavier and more demanding in terms of resources than our lighter kernel-based container proposal. In fact, each Docker container runs in the user-space, which increases both the system overhead and the activation time of that container. Alternatively, Linux containers (LXC) running in the kernel offer more agile bootstrap and operation than the ones provided by Docker containers. It has been proven that LXC is the most performant solution in nearly all virtualization scenarios [21].

The authors in [22] substantiated the importance of using more performant containers at the edge. However, they did not consider SDN. Recent work, such as [7,26], added a comprehensive literature discussion on how emerging IoT services can be enhanced through the collaborative deployment of both SDN and edge computing. To provide a solution that addresses the different challenges present in IoT environments, the current work proposes a novel integration of SDN with on-demand activation and orchestration of lightweight containers, some of which can be virtualized sensors.

The work in [23] investigated a resource provisioning framework for IoT applications in fog environments. This framework makes decisions empowered by Bayesian learning to enhance the latency and cost requisites of IoT services. The authors in [24] proposed an opportunistic admission and resource allocation policy to reconcile the uncertainty associated to distinct system aspects, such as (i) users requesting IoT services provided by vertical slices, (ii) the available system resources, and (iii) the heterogeneity level among types of system resources. The main idea of their proposal was to allow the service providers to decide whether to accept slice requests immediately or defer them according to the load and price of needed resources. The most intelligent offloading decision on either data or computation from edge devices with limited resources to other nearby resource-richer devices can aid in fulfilling the functional requisites of IoT-based services more efficiently [25]. The three last referenced contributions on resource provisioning did not directly consider the performance advantages of using the SDN paradigm.

The design and deployment of our proposal are discussed in the next section.

## 3. Proposed Solution

This section covers the proposal design in Section 3.1. The proposal deployment is explained in Section 3.2.

### 3.1. Architecture

Figure 1 presents the high-level design of our proposal. It is an FC system that combines the SDN paradigm with virtualized processing resources to be managed in IoT-based scenarios. Our proposal supports the management of edge-computing resources in elastic and agile ways. A possible distributed administration of edge storage is possible, eventually, in reaction to the evolution of spatio–temporal data popularity [27], or a future handling of flow mobility between different IoT access technologies could be enabled [28]. This is, however, beyond the scope of the present work, but we submit that a tight coupling of a SDN controller and a broker in edge nodes, such as middleboxes at user premises, are crucial for on-demand management of local heterogeneous resources [19].

In our proposal, the system’s data plane is formed by switches, which communicate with a controller through the southbound API. There is also a top-level broker system entity that communicates with the controller via the northbound API. In the beginning, not all containers in data plane are running, enhancing the system’s sustainability in a similar way to what was previously proposed for data center networks [29]. Then, the broker, after detecting an increase in IoT data traffic, can automatically boot additional containers to process that data.

Recently, various architectural standards and frameworks have been developed to address the issue of designing large-scale IoT networks. The core idea behind these architectures is to provide support for data, processes, and the functions of endpoint devices. There are many well-known architectures, such as oneM2M and the IoT world forum (IoTWF), and surveys are available that analyze the applicability of many reference architectures for the IoT [30,31] or even propose a simplified framework that emphasizes the basic components that are commonly found in most IoT systems [32].

Our solution is consistent with the shared understanding among these frameworks by using a multi-tier architecture to allow for the interconnection of the IoT endpoint de-vices to a network that transports the data to where they are ultimately used by applications, whether at the data center, in the cloud, or at the network edge, embedded in edge devices or middleboxes. Figure 1 depicts a horizontal split between two operation domains for networking and computation, similar to that suggested in [32]. Here, a broker, possibly controlled by IoT applications, is responsible for the service delivery and orchestration of networking and computation resources at the edge (bottom layers), which are made available, respectively, through SDN controllers and container hypervisors (middle layers). The networking domain resources are switches and links, while the computation domain resources are servers to process IoT data.

Considering the just-presented design, we also integrated the broker functionality into network functions inside the SDN controller, reducing the system complexity and eliminating the extra communication delay between the broker and controller. Additionally, a new top-level entity with the role of orchestrator among redundant SDN controllers was included in the system. These multiple orchestrated controllers can efficiently handle high control workloads and provide a more robust control plane against any potential failures or system threats [33,34]. The functions running in SDN controllers can automatically manage both networking and computational resources, resulting in a more efficient system operation.

The diagram in Figure 2 shows the most relevant sequential steps encompassing the diverse entities and how they interact in the investigated scenario.

The top-level manager depicted in Figure 2 orchestrates all the available controllers, assigning each controller the correct control OpenFlow role. In the figure, the cluster manager assigns the MASTER role to the only available controller [34]. We also consider a group of data servers (i.e., Server 1 to Server n) organized into an edge server cluster, which offers the same service to potential clients. Each edge server is initially idle, with its network interface switched off, in an attempt to preserve service resources, e.g., to diminish the energy consumption. When a client requests an edge service, the controller elects an edge server to attend to that client request. Before reporting back the appropriate server to the client, the controller activates the elected server, including its network interface. At this step, the controller uses the algebraic representation (NetworkX) of the discovered network topology to obtain a list of potential future end-to-end network routing paths (not shown in Figure 2). These paths can be immediately configured at the network nodes by the controller, prior to the normal activation of data flows. This controller behavior can help protect the traffic quality of long-time data flows. Returning to Figure 2, during the client/server session, there is the expected bidirectional exchange of data service messages. Finally, when the controller is notified of the end of the service session, it switches off the edge server.

### 3.2. Deployment

Figure 3 details the deployment of our proposal, which was introduced in Section 3.1. This proposal was implemented using a Ryu SDN controller, the broker, a virtual multilayer switch, and service containers. The main novelty of our work, in relation to the already existing functionality offered by Ryu APIs, is associated to the edge resource broker, which is either external to the Ryu controller (see Section 3.2.2) or embedded in the Ryu controller (see Section 3.2.3). In Section 3.2.2, the external broker complements the Ryu controller by identifying the need to boot service containers that will process IoT data. In Section 3.2.3, the embedded broker further enhances the Ryu operation by not only booting service containers but also later deactivating some of these containers in case they are no longer needed, thus releasing the normally scarce computational edge resources. Additionally, the Ryu controller was enhanced to support the single-step end-to-end proactive installation of OpenFlow rules at the network switches to improve the quality protection of new data flows associated with IoT edge services.

The implementation of Figure 3 materializes a testbed to obtain performance and functional results for evaluating the proposed solution. It uses a Ryu SDN controller and two Python applications to study distinct controller behaviors, i.e., reactive and proactive modes. Regarding containerization, Linux containers were chosen as a more lightweight and agile option compared to other alternatives, such as the Docker. The performance advantages of using Linux containers occur because they run directly inside the kernel. In the above scenario, the LXC1 and LXC2 containers virtualize IoT sensors and, as already mentioned, these containers are not always running. The virtual multilayer switch is an Open vSwitch (OVS), controlled by the Ryu SDN controller. The SDN controller uses Websocket northBound communication with the broker, which is responsible for managing the container’s lifecycle. All the mentioned components are installed in a single Linux machine, emulating a possible edge nodes setup implemented in middleboxes or set-top boxes where available IoT functions should be managed.

In more detail, the network topology, i.e., the data plane architecture, includes one Open vSwitch (OVS s1), one host (h1), and various links/bridges. Host h1 was created using Linux Namespace. The OVS s1 was configured with a static datapath-id value of one, the OpenFlow protocol version is 1.3, and a logical transport connection to the Ryu controller is in TCP port 6633, which is the default port used by the SDN controller. Further details about this are provided in the next subsection.

#### 3.2.1. Switch and SDN Controller Communication

As described above, two Ryu Python applications were deployed in the reactive mode. When the Ryu controller is started, a flow miss default rule is installed into the switch. In this way, the switch sends a PacketIn message to the controller whenever the switch (OVS) receives a packet. In reaction to each received PacketIn, the Ryu application uses the OFPActionOutput class and the OFPP_FLOOD flag in the PacketOut message, instructing the switch to forward the received packet to all switch ports except the incoming port.

The proactive Ryu application installs into the switch not only the “ask to the controller” rule but also rules that define the correct output port to forward the flow to a pre-defined Linux container. This proactive behavior reduces the number of PacketIn/PacketOut messages between the OVS and the controller, especially for elephant flows. Alternatively, the reactive mode may be better than the proactive mode for controlling sporadic and short-duration IoT data flows. In this way, the reactive mode reduces memory usage on switches with significant limitations in computational resources. The next subsection provides some information on how the Ryu controller and the broker communicate.

#### 3.2.2. Communication between the SDN Controller and the Broker

Whenever the SDN controller receives a packet from the switch, it sends a copy of the PacketIn header, in JSON format, to the broker via a Websocket connection, as seen in Figure 4. The broker then analyzes the header fields and extracts useful information, namely the physical and IP addresses, the transport protocol, and its ports.

Each Linux container has known IP and physical addresses. When an ARP message is identified, and the destination IP address matches one manageable Linux container, the broker can automatically boot that container. If the container is already running or if the destination IP does not match any of the pre-defined containers, the system logs corresponding messages for future analysis. This allows, for instance, the registration of the communication activity of individual virtual sensors and the deactivation of dormant containers. The next subsection discusses the novel automatic (de)activation of edge servers using SDN proactive flow control.

#### 3.2.3. Novel Function for Automatic (De)Activation of Edge Servers

The following text details how the novel controller functionality responsible for supporting the automatic deactivation of each edge cluster data server was implemented. Additionally, when the edge server goes into the sleeping state, the current network function is also responsible for activating that server when it is needed again. Figure 5 visualizes the timeline of system actions that we discuss below. The first message sent by the client device to the network is an ARP request to discover the MAC address of the data server to fulfill the client service request. This ARP request arrives at the ingress switch, and the switch sends a copy of it to the SDN controller, using a PacketIn OpenFlow message (see the leftmost action in Figure 5 and Algorithm 1 below). The controller, inside the function associated with the PacketIn Event, parses the ARP header and learns the Virtual_IP address of the cluster edge server previously indicated by the client. At this stage, the controller assumes the role of a Proxy-ARP and invokes the internal function designated as generate_arp_reply() (step 4 of Algorithm 1). When this function is invoked, it receives two arguments: the client IP and the client MAC. Then, the generate_arp_reply() function calls other function, boot_analysis(), passing two arguments: Virtual_IP and the client IP (step 18). This function selects a dedicated server for that specific client (step 23) and boots up the network interface of that server (step 24) before returning to the calling function (step 25). Next, generate_arp_reply prepares the ARP reply message (step 19), considering the returned IP of the elected server, and returns it to the function that is processing the PacketIn Event together with the MAC_server (step 20). At this step, our solution tries to optimize network operation. Thus, the controller holds the ARP reply (step 4) and checks if it is necessary to install a proactive flow rule (step 5). If proactive flow rule installation is necessary, the controller uses the algebraic representation of the network (NetworkX) to discover the end-to-end path with the minimum cost between the current ingress switch (of the ARP Request) and the elected server (step 6), which may include several switches. After this, for each switch in the discovered path (step 7), the controller executes the next proactive tasks: (i) it finds the ingress port and egress port in the minimum cost path (step 8) and (ii) it creates and sends the flow rules for the switch (step 9). These flow rules support both IP and ARP unicast bidirectional traffic, which are expected to traverse the path. In this implementation, we should be aware that in the case of elephant flows, the ARP protocol can send several verification ARP requests to prevent any potential ARP table-poisoning attacks. Therefore, after initially installing the flow rules in the path switches, the controller needs to keep track of that event, e.g., by using a Boolean variable (step 11), which is set to False. This variable will be set to True again after the flow rules become idle and are deleted from the switches, and the associated server will have its network interface switched off (see step 18 of Algorithm 2 below). The main advantage of using this Boolean variable is that it helps the controller handle ARP requests that follow the first one. In this scenario, the controller only confirms the MAC address of the elected server without installing new flow rules. Otherwise, repeatedly installing flow rules in the switches during a running session could negatively impact service quality. It is important to note that end-to-end paths for distinct sessions may be completely different, enabling load balancing. When the initial flows are fully installed at the switches, the controller can finally send the ARP reply to the client with the MAC address of the elected server (step 13).
**Algorithm 1** The controller processes the initial ARP request associated to each flow. That message is used as a trigger to anticipate the bidirectional flow rules installation on all the involved switches in the end-to-end flow path with the minimum cost between the client and the elected server.1:**for each** PacketIn Event with pkt do2: **if** pkt.ether.type = ARP3: **if** ARP_Request for discovering IP associated to Virtual_IP4:  MAC_server, ARP_reply_packet = **generate_arp_reply**(Virtual_IP, client IP, client MAC)5:  **if** install_flows6:  evaluate min cost path (ci) from current sw to server; store each switch si on that path7:  **for** each stored si do8:   finds out si ingress port & si egress port for the min cost path9:   creates and sends the flow rules to switch si10:  **end for**
11:  install_flows = False (see step 21 of Algorithm 2 below)12:  **end if**
13:  send_msg(current sw, PacketOut(ARP_reply_packet))14: **end if**
15: **end if**
16:**end for**17:**generate_arp_reply**(Virtual_IP, client IP, client MAC)18: IP_server = **boot_analysis**(Virtual_IP, client IP)19: Using the IP_server serializes the ARP reply message (pkt_arp_reply)20: **return MAC_server, pkt_arp_reply**
21:**end function**22:**boot_analysis**(Virtual_IP, client IP)23: selects a dedicated server for the client24: boots UP the network interface of server25: **return** IP address of the selected server26:**end function**

Now, let us analyze how Algorithm 1 impacts system performance when a new random data flow arrives in a network topology with *N* switches supervised by a single SDN controller. By analyzing the behavior of Algorithm 1, we can conclude that there are two positive outcomes for system performance, compared to the case when a non-proactive controller is used. When the controller behaves proactively, the first positive outcome for system performance arises due to a reduction in the number of OpenFlow control messages, i.e., PacketIn and PacketOut message types, compared to the situation with a non-proactive controller. This message reduction is represented by (1) and depends on *N*.
6 × *N* − 2(1)

The second positive outcome of using Algorithm 1 is due to the reduction in the number of network messages, e.g., ARP messages. This occurs because the controller assumes the role of a Proxy-ARP when it receives the initial ARP request from the ingress switch. The reduction of network messages is represented by (2); it also depends on *N*.
2 × *N* − 2(2)

The average trend of these two reductions in the number of messages per each new data flow is visualized in Figure 6. We simulated each topology size (depending on *N*) ten times. In each evaluation round, there was a random uniform probability of a new data flow being admitted to the network. Overall, the reduction in system messages is more significant for larger networks, especially for the control channel. Below, we discuss the behavior of our proactive controller after an IoT data flow has terminated.

To handle the scenario when a session between the client and the data server ends, the switch flow rules have idle timers. Thus, after the expiration of any flow idle timer in a specific switch, it sends an event-triggered OpenFlow message that notifies the controller about the removal of the flow (see Algorithm 2, step 7). Utilizing these notifications (steps 9–11), the controller (using the boot_analysis() function, which is called once every second (step 3); T = 1 s for step 4) turns off the network interface of the data server (step 17) and updates the state of a global internal variable (step 18), preparing the controller to initiate a future new functional cycle for switching on, keeping on, and switching off the network interface of the same data server. Therefore, the server remains operational for a few seconds after the service session ends. The controller also maintains useful statistical information, such as the total time each server was powered on, which can be used as accounting data to support a business model associated with the offered service.
**Algorithm 2** The controller can process any flow rule deleted message sent from any switch. Using the match fields of the deleted flow rules, the controller can identify the servers to switch off their network interface, saving server resources.1:**def** _**monitor**(self):2: **while** True do3: **boot_analysis**()4: hub.sleep(T)5: **end while**6:**end function**7:**for each** FlowRemoved Event do8: dpid = Event.msg.datapath.id9: **if** dpid == egress switch for data server cluster10: using match fields of removed flow rule, identifies the idle server11: memorizes the idle server to power off its network interface12: **end if**13:**end for**14:**def boot_analysis**(self):15: Verifies if each active server (servi) has a pending request to switch off its interface16: **for each** servi do17:  switches off the network interface of servi18:  install_flows = True (see step 11 of Algorithm 1)19:  **end for**20:**end function**

## 4. Evaluation Results and Their Discussion

This section presents the performance and functional tests carried out on the implemented proposal and discusses the most relevant results obtained. Figure 7 shows the initial testbed, which consists of a network topology formed by a software-based switch (OVS s1), a host (h1), and several containers (LXC1, LXC3). In addition, high-level entities such as the Ryu SDN Controller coupled with a broker manage the resources of the network topology. Table 2 lists the tools used during the proposal tests.

Section 4.1 discusses the performance results obtained from the initial testbed (Figure 7), regarding network configuration and container activation. Section 4.2 analyzes how our proposal behaves in an evolved scenario (Figure 8), subjecting our system to stress tests. Finally, again using the scenario of Figure 8, we present in Section 4.3 the evaluation results of the novel automatic (de)activation solution of edge servers using a proactive flow controller.

### 4.1. Delay Tests

The current subsection discusses evaluation results to assess the time required by our SDN system to perform the next two sequential functional steps: (i) powering on an initially disconnected service container; and (ii) sending the flow rules to network nodes that enable a distributed IoT data session between producer entities (e.g., sensors) and consumer entities (e.g., monitoring stations). These functional aspects under study are particularly notable to demonstrate the feasibility of on-demand managing resources that run on constrained devices at the network edge, such as small form factor single-board computers.

We tested two distinct operation modes of the SDN controller—reactive and proactive. In each operation mode, delay communication results were obtained in two scenarios: (i) communication between a host device, using the Linux network namespace and a sensor container; and (ii) communication between a user space container client and the same sensor container. The results for each scenario were obtained by sending four ICMP packets. In the first scenario, the sensor container (LXC1) sends ICMP packets to host h1. Alternatively, in the second scenario, ICMP traffic is exchanged between LXC1 and the container LXC3. Table 3 lists the average values from fifty runs of each testing scenario. As expected, the first packet always takes longer due to container activation delay. Comparing the results of both scenarios, scenario 1 has better response times than scenario 2. This is because in scenario 1, the kernel namespace h1 (i.e., Host 1) runs faster than the user space container LXC3 used in scenario 2.

In general, the worst response time results were obtained with the reactive mode. Although it is a simpler and less complex control mode, it stores only the default miss-flow rule in the switch flow table, which sends every received packet by the switch to the controller. This results in increased overhead on the OpenFlow control channel and overloads both the SDN controller and broker. As a result of these issues caused by the reactive mode, the system operates in a slower way, which increases end-to-end communication latency.

The above results demonstrate that even when the SDN controller and the controlled switch share the same computational space, such as edge middleboxes, the reactive mode operation, unfortunately, still introduces a considerable delay that can degrade the quality of most types of IoT applications. For this reason, the reactive mode may only be acceptable in certain situations, such as when the network needs to be highly dynamic and able to adapt quickly to changing conditions or with delay tolerant applications. The obtained results also show that the proactive mode, with a slight increase in configuration complexity at the network edge, can provide better performance results when long-time data sessions are active. However, this mode will not be able to adequately accommodate more volatile network conditions. The next subsection presents and discusses the results of stress functional tests carried out on the proposed solution.

### 4.2. Stress Functional Tests

The proposal has been tested further using more demanding scenarios (see Figure 8). In this realistic scenario, a TCP flow with 500 Kbit/s rate using Cubic congestion control was initiated in host h1 and sent to h4. Before establishing this communication, the proposal powers on switch s2, host h4, and enables the communication links s1–s2 and s2–h4, as summarized in Table 4, 1 Flow(A): h1–h4. This table also lists the major setup steps for additional stress tests, with an increasing number of simultaneous TCP flows.

The major performance results of these tests are visualized in Table 5. Delay1 is the average time interval between the ARP request and the ARP reply message of each end-to-end communication pair. Delay2 is the average time interval between the SYN message and the SYN-ACK message of each TCP logical connection. Comparing the obtained Delay1 results during tests 1 Flow(A) and 1 Flow(B), one can conclude that the time to power on both network and computational resources is about 1063 ms and the time to activate only the computational resources associated with the virtualized server h5 and connect it to the network infrastructure is much smaller, around 366 ms. In addition, when controlling a gradually increasing number of TCP flows, the solution showed a scalable and acceptable performance degradation, even during the highest delay of the operational period of the system (i.e., Delay1 in Table 5). The next subsection discusses the evaluation of the novel automatic (de)activation of edge servers.

### 4.3. Novel Function for Automatic (De)Activation of Edge Servers

The current subsection presents and discusses the evaluation results of the novel controller functionality that enables automatic (de)activation of the network interface of each edge server. The testbed used for this scenario is visualized in Figure 8. This scenario assumes that several services are demanded by clients h1, h2, and h3 from edge servers h4, h5, and h6.

The mgen was used as a traffic generator in each test, and the trpr was used to generate trend figures from all received messages that were previously stored in a log file. The following extract of used script lines shows how to start a test and create its log file (z = {4, 5, 6}; y = {1, 2, 3}).


*sudo ip netns exec hz mgen input rx.mgn output hz.log &*

*sudo ip netns exec hy mgen input tx_hy.mgn &*


The content of the mgen file that receives UDP flows at specified ports is as follows.


*# File rx.mgn*

*0.0 LISTEN UDP 30,000, 30,001, 30,002*


The contents of mgen files that generate UDP traffic with a Poisson distribution are presented below. For example, file tx_h1.mgn (when y = 1) initiates an 1 Mb/s UDP flow from client h1 to the data server h4 at time 10.0 s. This flow ends at the time instant 60.0 s.


*# File tx_hy.mgn (y = {1,2,3})*

*# Flow with 1 Mb/s between client hy and data server hz (z = {4, 5, 6})*

*10.0 ON 1 UDP SRC 20,000 + (y − 1) DST 10.0.0.89/30,000 + (y − 1) POISSON [976.75 128]*

*60.0 OFF 1*


To visualize the results of flows regarding rate, interarrival delay, and loss, the following commands were used.


*# Produce graphs for flow with 1 Mb/s between client hy (y = {1, 2, 3}) and data server hz (z = {4, 5, 6})*

*trpr drec input hz.log auto X output hz_rate.plt*

*trpr drec interarrival input hz.log auto X output hz_int_arrival.plt*

*trpr drec loss input hz.log auto X output hz_loss.plt*


To test the relevance of the controller in installing flows in the switches before they traverse the network, two tests were conducted. The first test used a proactive controller, and the second used a reactive controller. The differences in the Python code between the reactive and proactive controller are that, in the latter controller, inside the function that processes each PacketIn event, the code part that processes each ARP request to Virtual_IP (the IP address of the data server cluster, as described in Section 3.2.3) includes instructions for performing proactive discovery of the flow path with minimum cost and the immediate transfer of the flow rules to the switches, enabling the discovery of the optimum flow path.

The results of both tests are shown in Figure 9. The left part of the figure shows the results of the proactive controller, and the right part shows the results of the reactive controller. All the results are related to the flow associated with the service session between client h1 and data server h4. For each controller type, each visualized result is the average of the obtained values from five repetitions of the same test. When we compare the results for the proactive and reactive controllers, we can conclude that the reactive controller offers the worst results in the three flow performance metrics under study: rate, interarrival packet delay, and packet loss percentage. This degradation in flow performance occurs because when the controller operates reactively, both the controller and the switches become highly congested. The switches become congested because each switch must transfer many packets from the fast kernel datapath to the slow switch process in the user space before sending the corresponding PacketIn messages to the controller. This results in a significant degradation in switch performance. The controller also becomes congested because it is flooded by a huge number of PacketIn messages to be processed. All these system inefficiencies, when the controller operates reactively, diminish the quality of high-rate flows, due to switch congestion and packet buffering. In fact, when Figure 9a,b are compared, the service session controlled by the reactive controller is extended by 16 s from its initially programmed 50 s duration. This 32% extension on session time forces a high consumption of system resources, including considerable extra usage of computational resources associated with the edge server. Additionally, the maximum rate when the system is managed by a reactive controller is 15% ((1100–939)/1100) lower than the maximum rate obtained by the proactive controller. With a similar comparison for flow interarrival delay and flow loss percentage, one can conclude that the proactive controller (in relation to the reactive one) offers a 20% (20–0) reduction in maximum flow loss percentage and a very significant 83% ((206–34)/206) reduction in maximum flow interarrival delay.

Considering the proactive flow controller, we measured the system time slots shown in Figure 5 in Section 3.2.3. The measured time slots (ms) are shown in Figure 10. By far, the highest time slot is T7, which is 10.3 s. This is because the idle timer of the used OpenFlow flow rules was configured for 10 s.

As discussed earlier, the current (proactive) controller can (de)activate the network interface of each data server, based on the client demand for a specific edge service. Table 6 summarizes how the proposed function for automatic (de)activation of edge servers distributed the clients’ service demands among the virtualized servers (i.e., nodes h4, h5, and h6, as visualized in Figure 8). In this test, each client was uniquely associated with a specific server. For example, client h1’s service requests were always directed to server h4, which then offered the edge service exclusively to that client.

Analyzing Table 6, during the total system operation interval of 1431.70 s, the service was active for only 760.01 s. For example, during this time, server h4 provided the service for 303.31 s, which corresponds to 39.9% of the total time the service was active. This implies that the system resources normally associated with server h4 were free for other purposes during 60.1% of the total time the service session was active. Similar conclusions can be drawn for the remaining service servers, i.e., h5 and h6. This type of resource management is particularly relevant in network edge scenarios with constrained computational and network resources.

## 5. Lessons Learned

In the current work, we investigated a novel design for a programmable solution to control an edge computing domain with embedded devices. In this design, we considered a single controller and a broker, and we obtained some promising results showing that it is possible to use scarce edge resources more efficiently while maintaining the quality of IoT data services.

Our proposed automatic resource management function could also be responsible for putting system nodes into sleeping mode and disconnecting parts of the network infrastructure. These are very useful system features for subsequent use cases. One example is the management of the networking domain of an operator, where similar solutions to the ones investigated here can enhance the operation of limited system resources during peak demands. A second example in which our proposal contributes is the scenario where the controller records the operation time of each edge server. Considering that each edge server is associated with a specific client, this functional characteristic enables an accounting of the duration time and the amount of used system resources during each client service session. This can support a dynamic business model for IoT scenarios.

## 6. Conclusions and Future Work

The paper presented an open-source programmable solution that can activate or deactivate heterogeneous virtualized resources for services provided at the edge of the network, targeting IoT-based cases or others with limited system processing resources. We conducted extensive tests on the deployed open-source programmable system. We analyzed a considerable number of results and learned some important lessons after carefully examining those results. In summary, we showed that a programmable system can optimize the usage of scarce edge resources and protect the quality of IoT data services.

In our opinion, the current study has some open issues that could be addressed in future research. The first opportunity is to carry out an extensive investigation, using a detailed system model, about the optimum provisioning of edge system resources, including energy efficiency. Other possible enhancements to the current work would be to evaluate the proposal in a real IoT testbed, using standard solutions for several IoT services that make use of protocols such as the Message Queuing Telemetry Transport (MQTT), the Constrained Application Protocol (CoAP), or others. Finally, research on programmable solutions [35] is envisioned to control real embedded devices and activate edge containers in various network edge domains, such as creating a federated system of IoT data microservices.

## Figures and Tables

**Figure 1 sensors-23-02762-f001:**
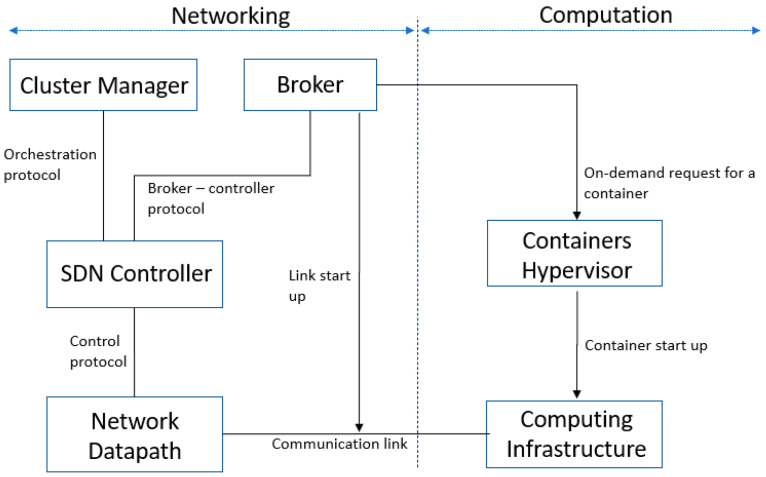
The proposed architecture with two operation domains, aggregating, respectively, networking and computation edge resources.

**Figure 2 sensors-23-02762-f002:**
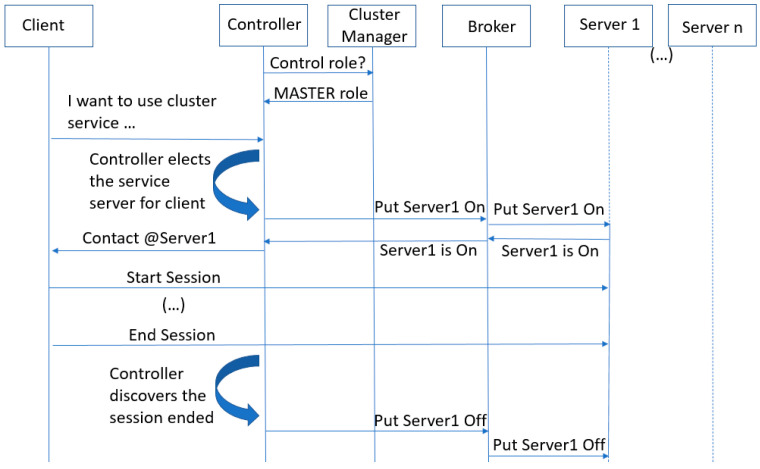
Sequential diagram with the diverse system entities.

**Figure 3 sensors-23-02762-f003:**
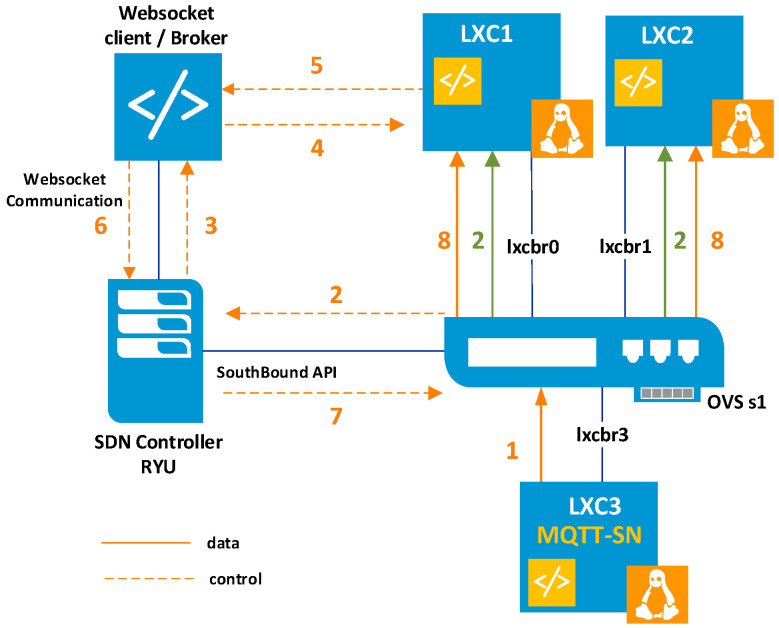
Deployed architecture (Cluster Manager is omitted). The visualized messages are numbered according to their relative order of occurrence in the system after a new data flow is admitted to the system, i.e., starting from the data message number 1.

**Figure 4 sensors-23-02762-f004:**
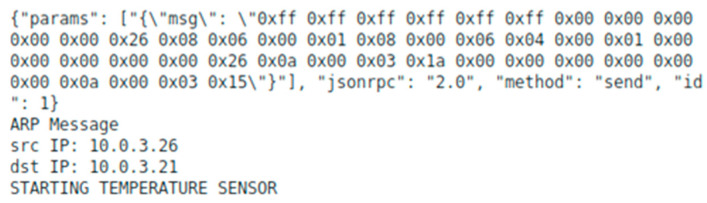
Exchanged message in JSON format from the controller to the broker.

**Figure 5 sensors-23-02762-f005:**
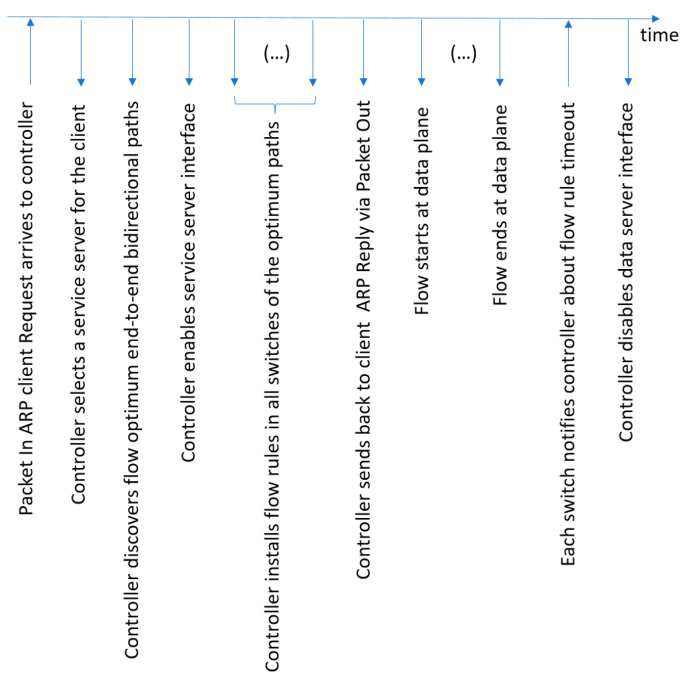
Timeline of system actions for the novel automatic (de)activation of edge servers using a proactive flow controller.

**Figure 6 sensors-23-02762-f006:**
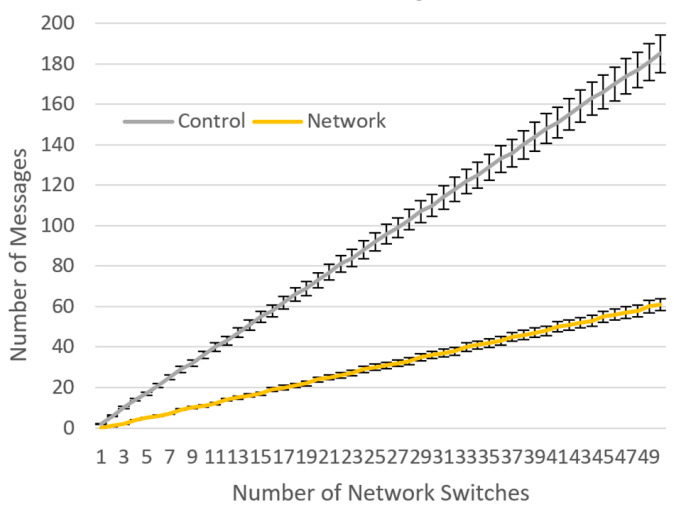
Average reduction in the number of messages caused by the operation of a proactive controller for each new data flow.

**Figure 7 sensors-23-02762-f007:**
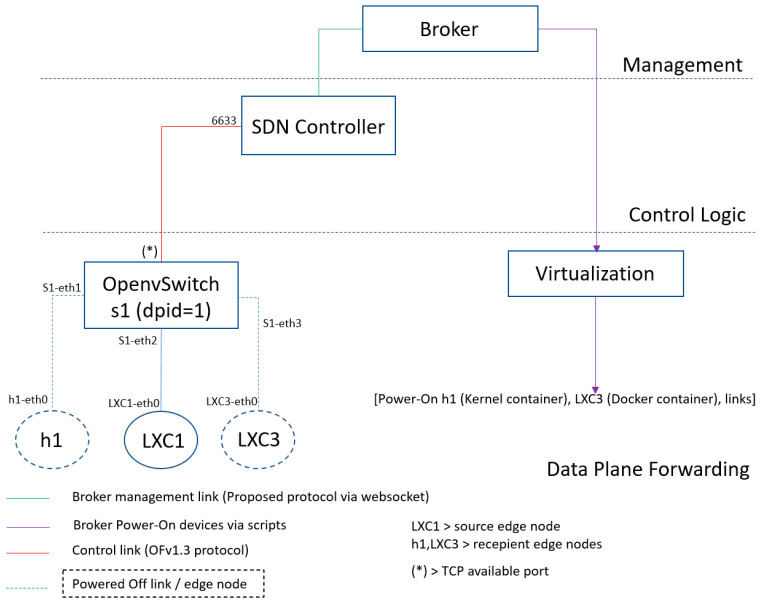
Initial testbed to evaluate the proposed solution.

**Figure 8 sensors-23-02762-f008:**
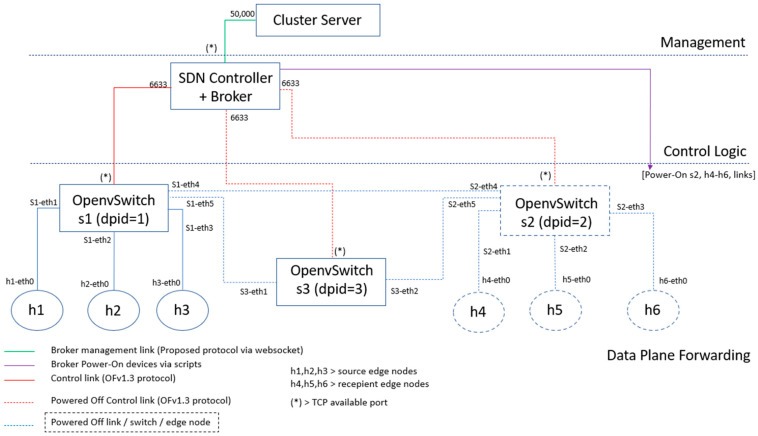
The testbed to study the system under stress and the novel (de)activation of edge servers.

**Figure 9 sensors-23-02762-f009:**
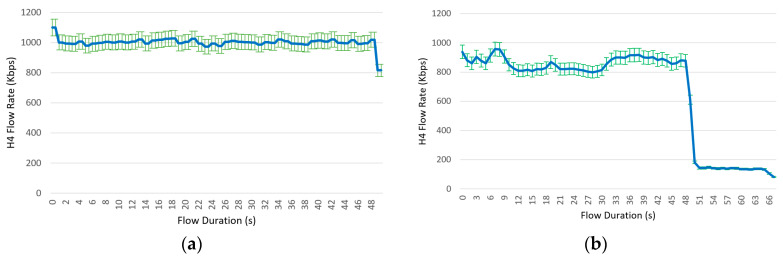
Average results of flow rate, flow interarrival packet delay, and flow loss percentage for the two compared controllers, during a 50 s service session between client h1 and data server h4. The proactive controller results are shown on the left side, and the reactive controller results are shown on the right. (**a**) Flow rate (Kb/s) for the proactive controller, (**b**) flow rate (Kb/s) for the reactive controller, (**c**) flow interarrival delay (ms) for the proactive controller, (**d**) flow interarrival delay (ms) for the reactive controller, (**e**) flow loss percentage for the proactive controller, (**f**) flow loss percentage for the reactive controller. In (**a**,**b**,**f**), the variation bars around non-null average results are visualized in green.

**Figure 10 sensors-23-02762-f010:**
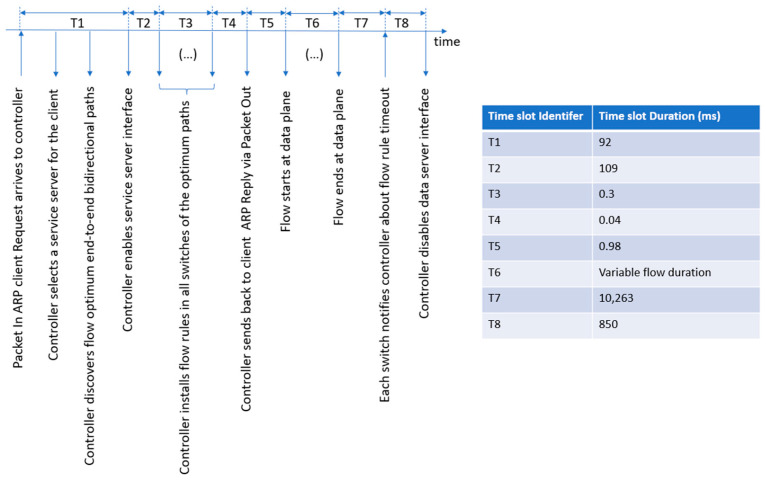
The temporal line of system actions associated with the novel automatic (de)activation of the edge server network interface using a proactive flow controller together, along with the measured time slots (in ms) between the most relevant system actions.

**Table 1 sensors-23-02762-t001:** Literature comparison of covered topics, marked by ‘+’.

Reference	Year	SDN	NFV (Broker)	ML	Orchestration	Uncertainty	Offloading	Network Slicing	Docker	Linux Containers	On-Demand Activation of IoT Services
[10]	2017	+		+							
[12]	2019	+	+	+		+					
[13]	2019	+		+							
[14]	2020	+		+							
[15]	2014	+	+		+						
[16]	2016	+	+					+			
[17]	2019	+	+								
[18]	2017	+					+				
[19]	1026	+				+			+		+
[20]	2012	+			+						
[21]	2018									+	
[22]	2019									+	
[23]	2020			+		+					
[24]	2022					+		+			
[25]	2021			+			+				
Our work	2023	+	+		+	+				+	+

**Table 2 sensors-23-02762-t002:** Hardware and software tools used during the evaluation tests.

ASUS Intel^®^ Core™ i7-3517U CPU @ 1.90 GHz 2.40 GHz, 12 GB RAM, Windows10 Education x64	-
VirtualBox Ubuntu 22.04	https://www.virtualbox.org/ (accessed on 29 January 2023) https://releases.ubuntu.com/22.04/ (accessed on 29 January 2023)
Ryu SDN Controller (v4.34)	https://ryu-sdn.org/ (accessed on 29 January 2023)
NetworkX (v2.6.3)	https://networkx.org/ (accessed on 29 January 2023)
OpenvSwitch (v2.16.90, DB Schema 8.3.0)	https://www.openvswitch.org/ (accessed on 29 January 2023)
Python 3.9.12	https://www.python.org/downloads/release/python-3912/ (accessed on 29 January 2023)
ip utility, iproute2–5.15.0, libbpf 0.4.0	https://man7.org/linux/man-pages/man8/ip.8.html (accessed on 29 January 2023)
Wireshark (v3.4.9)	https://www.wireshark.org/ (accessed on 29 January 2023)
Traffic generator MGEN (v5.02b)	https://github.com/USNavalResearchLaboratory/mgen (accessed on 29 January 2023)
TRPR (v2.1b11)	https://github.com/USNavalResearchLaboratory/trpr (accessed on 29 January 2023)

**Table 3 sensors-23-02762-t003:** Delay performance tests comparison, with 4 ICMP packets (values in ms obtained from Wireshark).

Controller’s Behaviour	PKT	Scenario 1 (Host 1 Is a Linux Container)	Scenario 2 (LXC3 Is a Docker Container)
		Host 1 -> LXC1	LXC3 -> LXC1
Reactive	1	416.56	565.94
	2	22.03	48.57
	3	16.84	34.04
	4	17.87	30.52
Proactive	1	201.70	521.66
	2	0.04	0.06
	3	0.04	0.07
	4	0.04	0.06

**Table 4 sensors-23-02762-t004:** Stress functional tests.

1 Flow (A): h1–h4	1 Flow (B): h1–h5	2 Flows: h1–h4; h1–h5	3 Flows: h1–h4; h1–h5; h1–h6	15 Flows: 5 × (h1–h4); 5 × (h1–h5); 5 × (h1–h6)
Power on s2 Power on h4 Connect s1–s2 Connect s2–h4 Inject 1 TCP flow of 500 Kb/s	Power on h5 Connect s2–h5 Inject 1 TCP flow of 500 Kb/s	Power on s2 Power on h4 Connect s1–s2 Connect s2–h4 Power on h5 Connect s2–h5 Inject 2 TCP flows of 500 Kb/s each	Power on s2 Power on h4 Connect s1–s2 Connect s2–h4 Power on h5 Connect s2–h5 Power on h6 Connect s2–h6 Inject 3 TCP flows of 500 Kb/s each	Power on s2 Power on h4 Connect s1–s2 Connect s2–h4 Power on h5 Connect s2–h5 Power on h6 Connect s2–h6 Inject 15 TCP flows of 100 Kb/s each

**Table 5 sensors-23-02762-t005:** Stress functional tests comparison (delay values in ms, obtained from Wireshark).

	1 Flow(A)	1 Flow(B)	2 Flows	3 Flows	15 Flows
Delay1	1063	366	2100	2072	2083
Delay2	18	15	25	54	135

**Table 6 sensors-23-02762-t006:** Time distribution of edge service session among the virtualized servers.

Server h4	303.31 s
	39.9%
Server h5	254.00 s
	33.4%
Server h6	202.70 s
	26.7%
Total Service Session	760.01 s (Service idle time = 673.25 s)
	100.0%

## Data Availability

Data sharing not applicable.

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
