# Peer review of "Elastic Provisioning of Network and Computing Resources at the Edge for IoT Services"

_sensors, 2023, doi:10.3390/s23052762_

Round 1

Reviewer 1 Report

This paper offers the design, implementation and evaluation of a programmable solution, which performs on-demand activation of offline IoT edge computing containers. This is achieved by using network functions at the high systme levels.

The paper is well written and presented. However, in my opinion, Section 3 should be improved:

- Section 3.1, where the proposed architecture is presented, must be improved, because it is not clearly presented. In figure 1 the correspondence between "networking" and "computation" must be better explained.

- Section 3.2. must also be improved. Here, authors must identify their contributions, with respect to the already existing APIs and interfaces in Ryu.

Reviewer 3 Report

The key contributions of this work should rewrite it, becasue they have some ambiguities.

Problem statement should be included within the introduction, and clearly list the contributions.

Sequential diagram with the diverse system entities. presented in the Figure 2 should be sync with the Figure. 1 and 3.

Nothing new learnt lesson if we consider the following text

After extensive tests made during the current investigation, it’s possible to discuss 503 the most important learned aspects. The first conclusion is in edge computing scenarios 504 with embedded devices and network devices controlled by system top-level programma- 505 ble functions, the proactive control mode is preferrable at the beginning of new flows in 506 detriment of alternatives, such as the reactive control mode. However, we should bear in 507 mind that the latter mode may adapt more quickly to volatile network states. 508 The usage of virtualization at the network periphery, for both network and compu- 509 tational resources offers a high flexibility and abstraction from the hardware heterogene- 510 ity and complexity. In addition, the virtualization enables an elastic (and hopefully cor- 511 rect) allocation of edge resources, following the variations on the demand for local ser- 512 vices, relying on those resources.

Authors have concluded the following text, but its part of implementaion

Additionally, we have also incorporated the functionality of the external broker in- 525 side the controller, to deploy and test a network function that after detecting that a specific 526 edge data server has no active flow destined to that server, this function switches off the 527 server network interface, for a possible liberation of additional server resources. But, later, 528 if the same server will be needed to attend for a new client request, its network interface 529 will be reactivated. This automatic resource management function could be also respon- 530 sible for putting in sleeping mode system nodes and even disconnecting parts of the net- 531 work infrastructure, which would be very useful system features in the next cases.

The entire Conclusions Section is full of trivialities and contains no findings, no pros and cons of this work and no future research directions. 

Round 2

Reviewer 1 Report

All my previous comments have been well addressed. The paper can be accepted in present form.

Reviewer 2 Report

Authors have made all the changes as suggested

Reviewer 3 Report

This paper may now be accepted due to the fact that the authors have done a good job of accommodating all our questions